# A Miniaturized Electrothermal-MEMS-Based Optical Coherence Tomography (OCT) Handheld Microscope

Qian Chen [1,2], Hui Zhao [1,3], Tingxiang Qi [2], Hua Wang [1] and Huikai Xie [1,2,3,*]

1   School of Integrated Circuits and Electronics, Beijing Institute of Technology, Beijing 100081, China; 7520220182@bit.edu.cn (Q.C.); 3220215101@bit.edu.cn (H.Z.)
2   BIT Chongqing Institute of Microelectronics and Microsystems, Chongqing 400000, China
3   LightVision Technologies Inc., Ltd., Foshan 528000, China
*   Correspondence: hk.xie@bit.edu.cn

**Abstract:** Swept-source optical coherence tomography (SS-OCT), benefiting from its high sensitivity, relatively large penetration depth, and non-contact and non-invasive imaging capability, is ideal for human skin imaging. However, limited by the size and performance of the reported optical galvanometer scanners, existing portable/handheld OCT probes are still bulky, which makes continuously handheld imaging difficult. Here, we reported a miniaturized electrothermal-MEMS-based SS-OCT microscope that only weighs about 25 g and has a cylinder with a diameter of 15 mm and a length of 40 mm. This MEMS-based handheld imaging probe can achieve a lateral resolution of 25 µm, a 3D imaging time of 5 s, a penetration depth of up to 3.3 mm, and an effective imaging field of view (FOV) of $3 \times 3$ mm$^2$. We have carried out both calibration plate and biological tissue imaging experiments to test the imaging performance of this microscope. OCT imaging of leaves, dragonfly, and human skin has been successfully obtained, showing the imaging performance and potential applications of this probe on human skin in the future.

**Keywords:** electrothermal MEMS; SS-OCT; handheld; human skin





## 1. Introduction

As the first protective barrier from the outside world, the skin is the largest organ of human body, and always closely responds to the balance of the endocrine environment [1,2]. The morphological structure of skin tissue is one of the important concerns that experts pay attention to regarding the physiological health of skin [3,4]. Existing clinical imaging techniques including high-frequency ultrasonography (US) and skin surface microscopy have been widely used for the diagnosis of numerous skin diseases, but still suffer from some limitations, such as poor contrast and a lack of depth information [5,6]. At the same time, benefiting from a high resolution, good imaging contrast and penetration depth, some other optical imaging modalities like laser speckle contrast imaging and photoacoustic microscopy have also been applied to study the vascular microcirculation of skin surface for studying the functional and structural information [7,8]. For example, Zhu et al. reported a transmission-type laser speckle contrast imaging technique for detecting blood flow distribution in thick tissue, successfully obtaining the information and dynamic changes in blood flow distribution in human subcutaneous skin with non-contact and a high resolution [9]. Yang et al. reported several innovative photoacoustic imaging techniques, such as 532/1064 nm dual-wavelength photoacoustic microscopy, a switchable optical and acoustic resolution photoacoustic dermoscope, and a miniaturized photoacoustic probe [10–12]. However, the resolution of laser speckle contrast imaging makes it hard to observe the capillary vasculature under a micro-level and still needs to be improved, and photoacoustic imaging always requires coupling gels, being easily affected by the bubbles inside the gels [13–15]. Featuring high optical sensitivity, a large penetration depth, non-contact and non-invasive imaging capability, optical coherence tomography (OCT)

is considered a promising technology for human skin imaging [16]. Fujimoto et al. and Fercher et al. successfully developed the first time-domain OCT (TD-OCT) and Fourier-domain OCT (FD-OCT) in 1991 and 2002, respectively [17,18]. With faster imaging speed and good sensitivity as there is no need to move the reference arm to obtain the depth information, FD-OCT attracts more attention. Utilizing different laser sources and detection methods, two FD-OCT techniques, spectra-domain OCT (SD-OCT) and swept-source OCT (SS-OCT) were developed and applied in dermatology [19,20]. For example, Anthony J. Deegan et al. used a self-made clinical prototype OCT system to acquire OCT/OCTA images of patients undergoing medium and thick skin grafts after severe skin burns at multiple time points and multiple locations [20]. Compared with SD-OCT, SS-OCT has larger bandwidth, deeper penetration of biological tissues, and a high instantaneous coherence of sweep light sources, allowing for deeper longitudinal imaging ranges. SS-OCT is suitable for real-time observation of skin tissue structures with high resolution, fast imaging speed and relatively large imaging depth [21].

Most of the existing SS-OCT systems usually utilize galvanometer scanners for laser scanning, with a size similar to that of a tabletop optical microscope [22,23]. Handheld SS-OCT probes were proposed, but they were still bulky and inconvenient for long-time handheld imaging [24–26]. Thus, it is necessary to find miniaturized fast scanners for handheld SS-OCT imaging [27]. Micro-electro-mechanical system (MEMS) micromirrors are small and fast, and can thus be used to reduce the size and weight of imaging probes in optical imaging systems [28]. Many kinds of MEMS-based imaging techniques have been reported and applied in human skin imaging, demonstrating the feasibility of realizing miniaturized, portable and handheld imaging [26–30].

In this study, we proposed a miniaturized electrothermal-MEMS-based SS-OCT microscope, with a small size and light weight for long-time handheld clinical use. We applied the probe to capture the structure of some biological samples, such as the veins of leaves and the wings of a dragonfly. We also employed it to observe the tissue structures of a volunteer's finger, fingernails and wounded skin, showing the clinical feasibility and potential of this technology.

## 2. Materials and Methods

### 2.1. System Configuration and the Imaging Probe Design

Figure 1 shows the configuration of the SS-OCT system (LVM-1000, Light Vision, China). A swept laser source (HSL-20, Santec, Japan) was utilized to emit a 1310 nm laser beam with a 100 nm bandwidth. The laser is transmitted through a single-mode fiber (SMF) and divided into two laser beams via a 2 × 2 coupler. One laser beam reaches the mirror in the reference arm through a polarization controller (PC) and a collimator (CL), while the other laser beam passes through the sample arm whose end is an MEMS-OCT probe. Interference signals are produced by the two laser beams reflected back from the reference arm and sample arm (imaging probe), respectively, and are detected via a balanced photodetector (PDB570C, Thorlabs, USA). Then, the signals are collected using a high-speed data acquisition card (ATS-9350, Alazar Inc., Canada) with a sampling rate of 250 MS/s. An MEMS controller is used to drive the MEMS mirror to generate 2D raster scanning and thus realize 3D OCT imaging. An external trigger signal is used to keep the synchronization among the laser emitting, MEMS scanning, and data acquisition.

In the bottom right corner of Figure 1, the schematic of the imaging probe is shown in the black dash box, with all optical and electrical components inside a small handheld tube. The optical path is designed as simply as possible, and a minimum number of components is used to reduce the volume size and weight of the probe. In detail, the laser beam output from the fiber is collimated via a fiber collimator (F230FC-C, Thorlabs Inc., USA), and then converged using a doublet lens with a diameter of 6 mm and a focal length of 30 mm (GCL-010601, Daheng Optics, China). An electrothermal MEMS scanner (WM-L5-5, WiO Tech., China) is used to reflect the converged light beam to the sample surface, and scans the laser beam in a raster pattern for three dimensional (3D) imaging. The miniaturized

handheld probe is a cylinder with a diameter of 15 mm and a length of 40 mm, with a weight of 25 g. It has a spatial resolution of 25 μm, a temporal resolution of 5 s, and an imaging FOV of $3 \times 3$ mm$^2$.

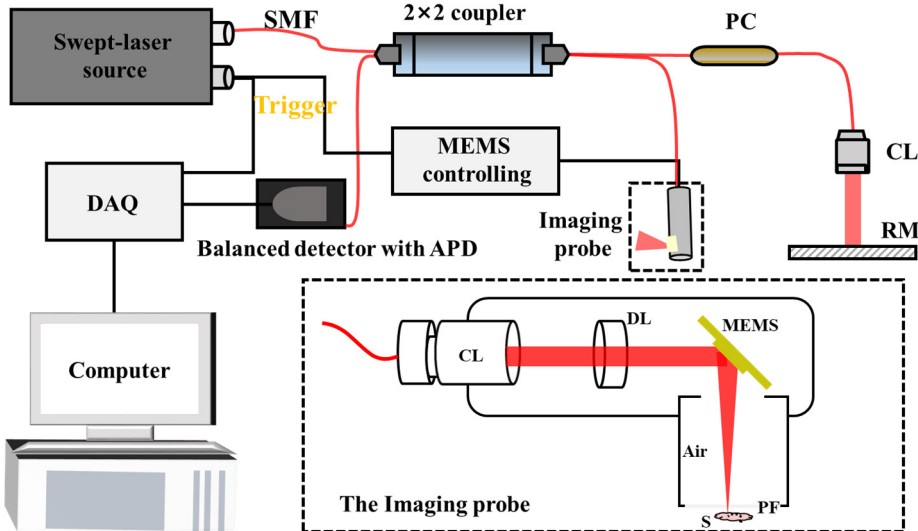

**Figure 1.** The schematic of the SS-OCT system and the inside structure of the imaging probe. SMF, Signal-Mode Fiber; PC, Polarization Controller; CL, Collimator; RM, Reflector Mirror; APD, Avalanche Photodiode; DL, Doublet Lens; PF, Polymer Film; S, Sample.

### 2.2. Sample Preparation and Experimental Process

The commercial and custom light stereotypes with target patterns are used to test the basic imaging performance of the imaging probe. In addition, two types of leaves with thick and dense veins were chosen and boiled in a 10% sodium hydroxide solution for 5 min. When the leaves turned black, they were rinsed carefully with clean water. A small test tube brush with soft hair was used to gently brush off the rotten mesophyll and clean it until only the veins were left. Finally, the leaf veins were dried to form experimental specimens for imaging. Dragonfly samples were purchased online, and the wings were partially used for biological imaging tests. For skin imaging, due to the characteristics of contactless, safe, and non-destructive imaging mode, after seeking the oral and written consent of a healthy volunteer, we used the probe to perform simple imaging operations on various parts of her skin, without involving any complex procedures. The single imaging time was about 5 s, and all skin imaging experiments were finished within a few minutes, which also verifies the advantages of portability and speed of skin imaging by using our proposed handheld probe.

### 2.3. Data Acquisition and Processing

Commercial software (LVM-1000, OCT Viewer, Light Vision, China) was used for data acquisition, processing, and image reconstruction. Within the software's platform, the underlying code was written based on Labview2019 software, with the functions of original signal filtering, Fourier transform processing, image reconstruction based on acquired 3D data, and image brightness enhancement, among others.

## 3. Results

### 3.1. Parameter Testing of the MEMS and MEMS-Based Handheld Probe

As the most important scanning components, a photograph of electrothermal MEMS inside the probe is also shown in Figure 2a. In Figure 2b,c, the testing curve of MEMS shows that the optical deflection angle is proportional to the driver voltages, and allowed for a maximum optical scanning angle of 8 degrees, with a 4 V DC voltage on the arms of MEMS, which corresponds to a maximal scanning range of 3 mm. We found that even though

the MEMS mirror deflection has a similar linear correspondence to the voltage, when the given voltage is smaller than 1.6 V, and between 1.6 V to 4 V, it still has slightly different slopes. This is related to the circuit characteristics of the MEMS chip and inconsistency of the resistance values of the four MEMS drive arms. This is a well-known phenomenon for electrothermal MEMS mirrors [31,32].

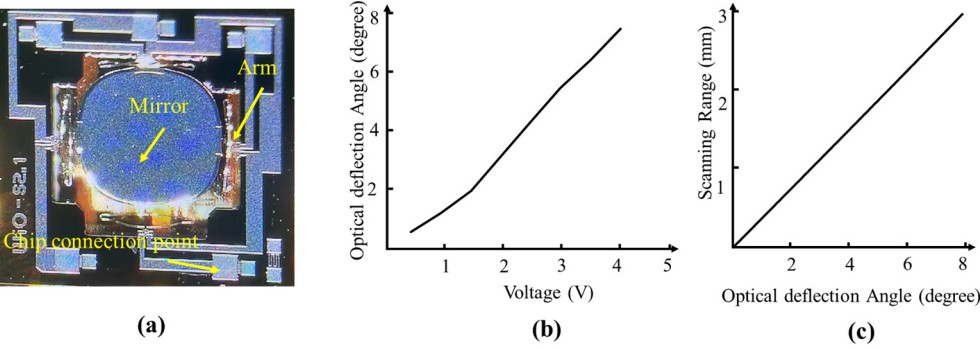

(a)                      (b)                      (c)

**Figure 2.** Photograph of MEMS and its performance testing. (**a**) Photograph of the MEMS under optical microscopy. (**b**) The testing optical deflection angle and driver voltage curve of MEMS. (**c**) The scanning range of the MEMS under different optical deflection angles.

A photograph of the probe is shown in Figure 3a, which is miniaturized and suitable for handheld imaging. To evaluate the spatial and lateral resolution of the SS-OCT probe, we imaged the pattern of a resolution plate. The minimum 5–3 line pair can be distinguished, indicating a practical imaging resolution of 25 μm, as shown in Figure 3b. We also tested the penetration depth of the probe by imaging the stack of cover glass with a known layer thickness of 150 μm. Figure 3c shows that the maximum number of observed cover glass is about 22, which corresponds to the penetration depth of 3.3 mm. In addition, we also imaged the photolithography patterns with known scales, and tested the FOV of the probe, which is about $3 \times 3$ mm$^2$, as shown in Figure 3d.

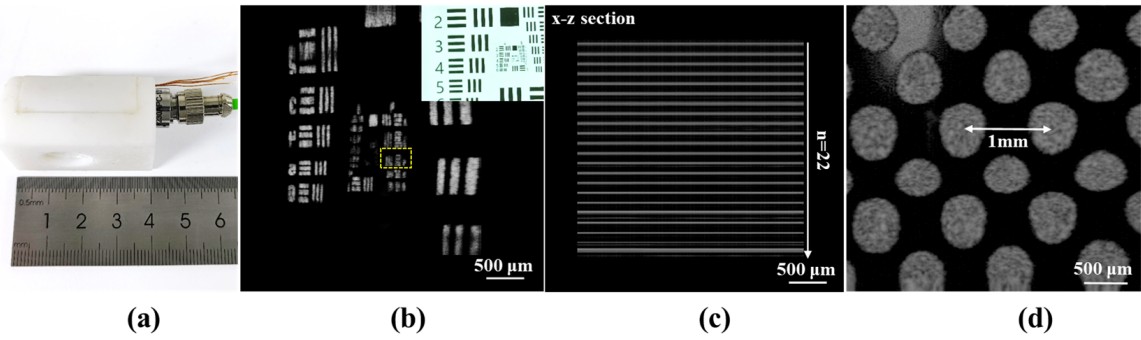

(a)                      (b)                      (c)                      (d)

**Figure 3.** Photograph of the handheld SS-OCT probe and its imaging testing including imaging resolution, penetration depth and FOV. (**a**) Photograph of the imaging probe. (**b**) Using the microscope to image the known resolution plate. (**c**) The imaging result of the biological cover glass sheet stack, with a known thickness of each glass sheet. (**d**) The imaging result of a custom mask plate with known photolithography patterns for FOV calculation.

### 3.2. The Ex Vivo Biological Sample and In Vivo Human Skin Imaging Results

To determine the performance of the miniaturized SS-OCT probe, we applied it to observe the biological specimens with dense vein structures. Figure 4a,b shows the imaging results of the veins' texture in two leaves, showing sparsely and densely reticular structures, respectively. In addition, we also captured the veins of dragonfly wings, which were quadrilateral or pentagonal grids, as shown in Figure 4c. The enlarged view of a localized dragonfly wing marked with a yellow dash box shows that the wings were filled with

complicated columnar structures, which played important roles during flight, helping to keep the wings from becoming stuck to each other with water and dust. These results demonstrated the potential of the SS-OCT microscope in high-resolution observation of microstructures of biological specimens.

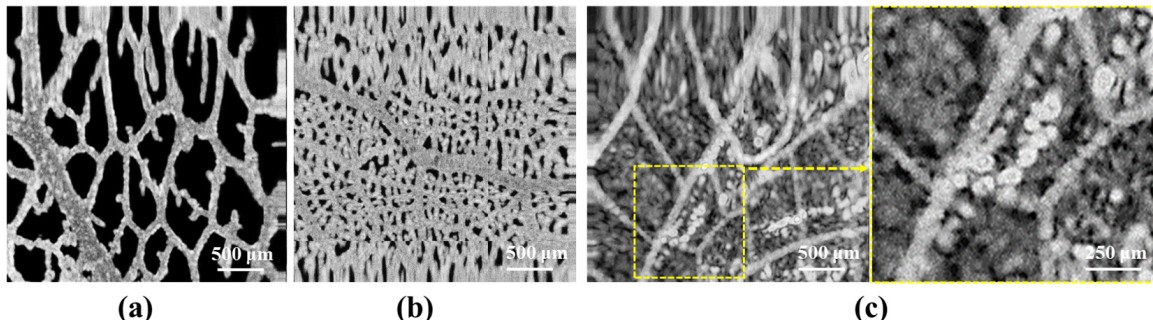

**Figure 4.** Using the SS-OCT microscope to observe the microstructure of biological specimens like leaves and dragonfly. (**a**,**b**) The sparse and dense vein structure of two strains of leaves. (**c**) The wing structure of a specimen of dragonfly and an enlarged view of a localized dragonfly wing marked with a yellow dash box.

We also carried out in vivo skin imaging experiments, including the fingernails, finger joints and the inner wrist, by using the miniaturized SS-OCT microscope. Figure 5(a-1) shows the photograph of a fingernail under biological microscopy. We carried out OCT imaging on the marked local site (red box), obtained the 3D visualization and x-z section view of the fingernail, and found densely and regularly connective tissue structures, as shown in Figure 5(a-2),(a-3). Furthermore, we employed the probe to observe the skin tissue structure and deep blood vessels on human finger joints, as shown in Figure 5(b-1)–(b-3). The skin tissue structure of the volunteers' inner wrist was visualized using OCT imaging, including the epidermal folds, internal blisters, and damaged skin tissue on the scalded site, as shown in Figure 5(c-1)–(c-3). In the soft skin tissue, obviously irregular boundaries due to the rich biological compositions were observed. The in vivo imaging results also showed that the miniaturized probe had the capability to visualize skin structures with satisfactory imaging performance.

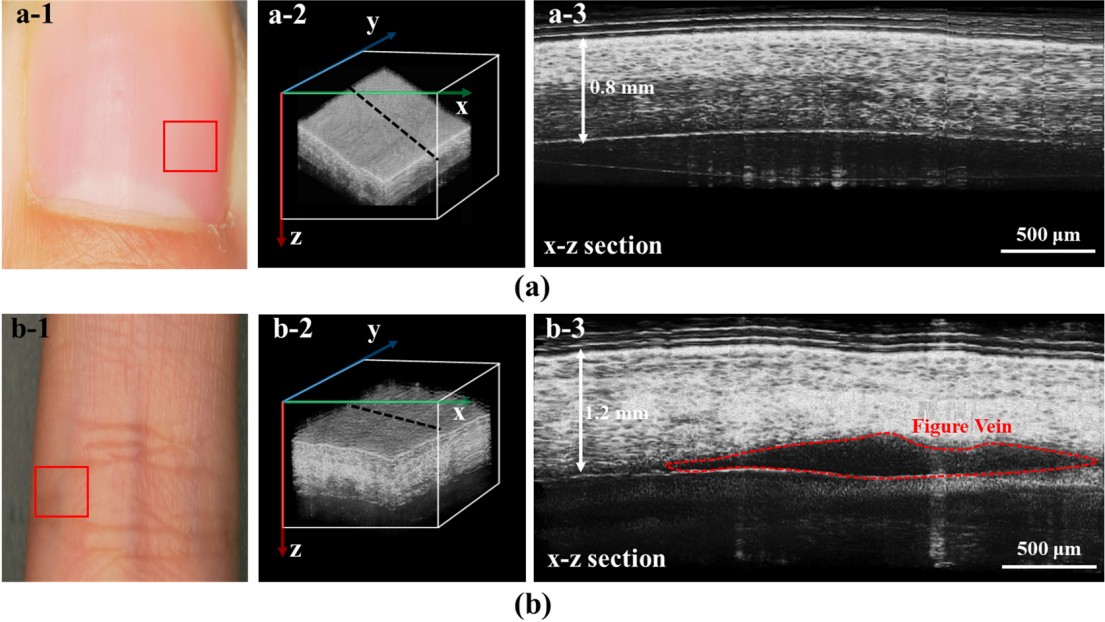

**Figure 5.** *Cont.*

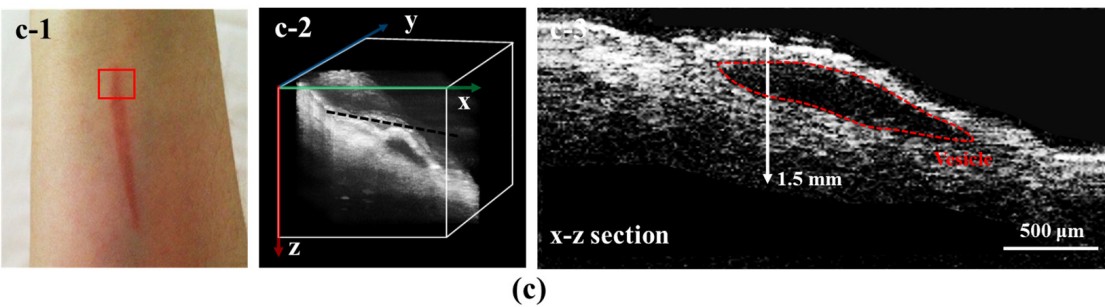

**Figure 5.** Photographs and corresponding SS-OCT imaging results of a fingernail (**a**), finger joint (**b**), and inner wrist skin (**c**). (**a-1**) The observation of a fingernail under biological microscopy. (**a-2,a-3**) The 3D visualization and x-z section view of the fingernail structure via SS-OCT imaging. (**b-1**) The observation of a finger joint under biological microscopy. (**b-2,b-3**) The 3D visualization and x-z section view of the skin structure in figure joints via SS-OCT imaging. (**c-1**) The observation of a volunteers' inner arm wrist under biological microscopy. (**c-2,c-3**) The 3D visualization and x-z section view of the localized scalded skin on a volunteer's inner wrist.

## 4. Discussion

SS-OCT is a popular imaging technology and is widely used in fundamental and clinical studies [33–35], especially for biological tissue imaging, considering that skin tissue has strong optical absorption and scattering characteristics, which will affect the penetration depth of lasers. In order to maximize the imaging depth, the near-infrared band is generally used for imaging, such as 1060 nm or 1310 nm. With an increase in the wavelength, the scattering decreases strongly. Therefore, we finally selected the 1310 nm wavelength to minimize the scattering even though 1310 nm has stronger absorption from the water than 1060 nm. While imaging, 3D imaging with high performance can be realized using either a motor stage or galvanometer. However, motor stages have a slow scanning speed and will cause motion artifacts, and this scanning method is gradually eliminated. On the other hand, the use of galvanometer scanning for imaging can solve the above problems, but still has a bulky size, which will lead to operation fatigue with handheld imaging for a long time [36,37]. With the rapid development of MEMS, electrothermal MEMS mirrors have been widely used in biological imaging, which benefits from their low-voltage driving, large scanning angle, small size and low cost [30,38]. In this study, we applied electrothermal MEMS to SS-OCT imaging, and demonstrated a handheld imaging probe with a simple optical design and high imaging performance, which greatly reduces the instrument's size and weight, and provides a new technology for biological imaging.

This paper describes a handheld SS-OCT probe with an internal simple optical light path and relative miniaturized mechanical structure that is easy to install. Using a 2D electrothermal MEMS for laser scanning and imaging can effectively reduce the size and weight of the probe, with a low cost. We tested its imaging performance including a lateral resolution of 25 μm, a penetration depth of 3.3 mm, and a scanning range of $3 \times 3$ mm$^2$, among other factors. In addition, we also used the handheld probe to carry out high-resolution imaging of in vitro biological samples and in vivo human skin tissue, demonstrating a good three-dimensional imaging performance. It is worth explaining that the theoretical penetration depth of a 1310 laser is large, but the actual biological imaging depth is still relatively shallow, which is closely related to the physical characteristics of different tissues. For example, the nail imaging depth is only of about 0.8 mm due to the thick interdigital cortex, and the skin tissue can only reach about 1.5 mm imaging depth because of the epidermal pigmentation and blood vessel absorption in the dermis, which may reduce the useful backscattered light (Figure 5(a-3),(b-3),(c-3)). The more tender skin tissue, such as the face skin and the inner arm skin, will allow for a more satisfactory imaging depth. Even so, the nail imaging results still clearly show the distribution of the stratum corneum structure, showing the potential to study gray-nail-related diseases. The human finger abdominal imaging results also demonstrate that the SS-OCT probe has

potential applications in arthritis studies. In addition to benefiting from the non-contact and non-invasive imaging capability, this MEMS-based SS-OCT microscope may have good potential applications in the dermatology of burn and cosmetic surgery.

It is worth considering that our proposed handheld MEMS-based microscope still has some limitations and needs further improvements. The relatively small size of the MEMS mirror plate may cause some optical loss and decreased penetration depth, while limiting the size of the spot irradiated will also lead to insufficient imaging resolution. Although an MEMS can achieve scanning with safe and low voltages, the scanning angle and the performance of resistance to vibration and drop damage still need to be improved. In the future, it is important to develop or use innovative MEMS scanners with a larger scanning angle and better anti-fall characteristics. We also need to optimize the internal design of the handheld microscope to improve the lateral resolution, FOV and imaging stability. The design of a user-friendly handheld appearance should also be considered to make it more practical and useful.

## 5. Conclusions

In summary, we developed a miniaturized electrothermal MEMS-based SS-OCT microscope probe, which has the characteristics of small size, light weight, high resolution and fast imaging speed. We carried out both in vitro biological specimen imaging and in vivo human skin imaging experiments. The imaging results showed that the microscope has the potential to examine biological tissue structures in human skin. Our proposed MEMS-based handheld SS-OCT microscope has a simple structure, which is more lightweight and portable to use. Within the probe, the scanning voltage of the MEMS mirror is also safer and more reliable than that of the existing benchtop or galvanoscope-based imaging terminals. In the future, we need to optimize the key scanning components of the MEMS probe to achieve a better imaging performance, such as higher resolution, larger FOV, a more user-friendly design and more comfortable use. We will also add more functional modules inside the system, combined with other imaging modes such as photoacoustic microscopy or confocal fluorescence microscopy, to obtain richer structural and functional information. These new techniques can be used for the diagnosis and treatment of clinic dermatoid diseases of scar hyperplasia, melanoma, pemphigus, acne, and psoriasis, among others. Meanwhile, we will also continue to use the proposed SS-OCT probe to carry out more profound biological application research to study some developmental processes and mechanisms of diseases.

**Author Contributions:** Five authors specify their individual contributions to this research article. Conceptualization, H.X.; methodology, H.X. and Q.C.; software, H.Z. and H.W.; validation, Q.C. and T.Q.; formal analysis, Q.C.; investigation, T.Q.; data curation, Q.C.; writing—original draft preparation, Q.C.; writing—review and editing, H.X. and Hui Zhao; visualization, Q.C.; project administration, H.X.; funding acquisition, Q.C. and H.Z. All authors have read and agreed to the published version of the manuscript.

**Funding:** This research was funded by the China Postdoctoral Science Foundation, grant number 2022M720439, and the Foshan Technology Innovation Team Project, grant number 2018IT100252.

**Data Availability Statement:** All the data and codes used to derive the findings of this study are available from the corresponding author upon reasonable request.

**Conflicts of Interest:** Authors, Hui Zhao and Huikai Xie were employed by the company The LightVision Technologies Inc. Ltd. The remaining authors declare that the research was conducted in the absence of any commercial or financial relationships that could be construed as a potential conflict of interest.

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
