# Peer review of "A Miniaturized Electrothermal-MEMS-Based Optical Coherence Tomography (OCT) Handheld Microscope"

_photonics, doi:10.3390/photonics11010017_

Round 1
Reviewer 1 Report
Comments and Suggestions for Authors
Please find my comments in the file that is attached here.

Please find my comments in the file that is attached here.
Author Response
We appreciate the valuable comments raised by the editor and reviewers. We have addressed all the concerns one by one, made a major revision of manuscript, and highlighted the revisions of the revised manuscript in red. (The revised manuscript can be seen in the attachment)
1. The authors didn't clarify that this manuscript is a follow-up of their previous paper referenced as number [15].
Response: Thanks for pointing this out. We have carefully rechecked all references, added and updated some literature to improve the manuscript. Please refer to Page 1, lines 40-42.
2. The introduction section should be improved.
Response: Thanks. We have improved the introduction section of the manuscript. Please refer to Page 1, lines 30-43 and Page 2, lines 46-57, 60-64, 66-68.
3. The detailed principle of operation of the device should be explained in section 2.
Response: Thanks for the reviewer’s suggestion. We have added section 2.1, section 2.2 and section 2.3 in the original section 2. In section 2.1, the detailed principle of operation of the device has been added for better understanding the SS-OCT imaging. Please refer to Page 2, lines 79-89 and Page 3, lines 107-127.
4. Figure 1 is crammed and not of good quality.
Response: Thanks for pointing this out. We redrew the system diagrams for better layout and image clarity, and replaced the previous Figure 1(a) (see Page 3, lines 101-106).
5. All figures throughout the manuscript are low quality and outside the paper's margins.
Response: Thanks for pointing this out. We have re-adjusted the clarity and size of the images for a more comfortable reading experience. (Page 3-6, figures 1-5)
6. Figure 2(a) is not clearly labelled. Why are there two slopes in Figure 2(b)?
Response: Thanks for the reviewer’s suggestion. We have added the annotations in figure 2(a). In figure 2(b), by testing the optical deflection Angle under different driving voltages. We found that the MEMS mirror deflection does not show a completely linear correspondence to the voltage within the driving voltage from 0V to 4V. When the given voltage is very small (<1.6V), its deflection angle characteristics are not obvious, while the given voltage is relatively large (1.6V-4V), it has a larger deflection angle and slope. This phenomenon may be related to the inconsistency between the actual drive voltage and the real drive voltage is given to the four arms of MEMS, and also may be related to the inconsistency of the resistance values of the four MEMS drive arms. We added the explanation in the manuscript. (Page 4, line 134-139)
7. The scale bars in Figure 5 are wrong. This suggests misleading results and conclusions.
Response: We are very sorry for the mistakes and thanks for the reviewer’s reminder. We have carefully re-checked the contents of Figure 5. The longitudinal imaging depth are right and accord with common sense, and we have modified the correct scale bars in the horizontal scale. (Page 5-6, line 188-190)
8. It is not clear what is the actual size of the device as it is 'miniaturized'.
Response: In the paper, we claimed that the front-end handheld imaging probe is miniaturized, but not the overall system miniaturization. In general, it is hard to realize a portable and miniaturized SS-OCT imaging system, due to the size limitations of some key equipment inside the system. However, the front-end imaging part can be designed to connect (plug or insert) to the whole machine for handheld imaging, making the imaging more portable and flexible. In this paper, only the hand-held probe size is described and considered to be miniaturized, and the overall system size is not described. The size of the integrated system (including the display screen ) is about 45×45×45 cm3. (Page 3, line 98-100
9. There are too many references, although they are not all used in writing the manuscript.
Response: Thanks for the reviewer’s suggestion. We have carefully rechecked all references, added more literature and updated some literature to improve the manuscript. (Page 8-9, references 1-38)
Reviewer 2 Report
Comments and Suggestions for Authors
The authors present a miniaturized swept-source OCT system based on an electrothermal-MEMS handheld cylindrical imaging probe that weighs only 25 grams, and has a diameter of 15 mm and length of 40 mm. The authors characterize the system’s performance in terms of lateral resolution, field-of-view and imaging depth and present images from leaves, dragonfly wings, and human skin.
The presented system is interesting, and the performance is convincing. The paper is well-written and well-organized. However, a clear claim of the innovation and originality of the presented work is missing. This is the main issue that must be addressed for the paper to be considered for publication.
In fact, the paper does not contain any comparison of the developed system against other similar OCT systems. There are several papers concerning handheld OCT systems for skin imaging, some of them using MEMS devices for lateral scanning (see for example Sancho-Durá et al., DOI: 10.1002/jbio.201800193). There are also available commercial handheld OCT systems for skin observation. So, the authors must discuss how their system compares with those already presented in the literature and/or commercially available and report exactly what is the novelty and originality of their system.
Other issues:
The authors should also discuss why they selected a wavelength of 1300 nm and not 1060 nm, also available in SS-OCT technology. This discussion should be made in terms of skin absorption and scattering.
Wouldn’t be preferable to use balanced detection instead of removing common-mode interference and DC terms by signal processing?
I suggest indicating in Figure 3b the 5-3-line pair of the USAF test target.
Comments on the Quality of English LanguageThe English usage is adequate.
There is a typo in line 83: “angle” instead of “angel”.
Author Response
We appreciate the valuable comments raised by the editor and reviewers. We have made a major revision of to improve the manuscript, addressed all the concerns, and highlighted the revisions of the revised manuscript in red.(The revised manuscript can be seen in the attachment)
1. The authors should also discuss why they selected a wavelength of 1310 nm and not 1060 nm, also available in SS-OCT technology. This discussion should be made in terms of skin absorption and scattering.
Response: Thanks for the reviewer’s suggestion. We have added some discussion about the skin absorption and scattering, and explain why we selected the wavelength of 1310 nm. (Page 6, line 200-206)
2. Wouldn’t be preferable to use balanced detection instead of removing common-mode interference and DC terms by signal processing?
Response: Thanks for the reviewer’s suggestion. In fact, we used a balanced detector with an avalanche photodiode inside to process the signal in the integrated system. However, we only give a simple modular classification description of the original system diagram in the manuscript. We have updated the system diagram and added a detailed description of the imaging principle. (Page 2, line 79-89)
3. I suggest indicating in Figure 3b the 5-3-line pair of the USAF test target.
Response: Thanks for the reviewer’s suggestion. We added the USAF test target (RTS3AB-P) in the figure 3(b) for an indication. The parameter mapping table of the test target is shown as follows. The resolution is calculated by 1000/40.3175=25 μm. We have revised the value in the manuscript. (Page 1, line 15-16; Page 3, line 99-100; Page 4, line 147-148; Page 7, line 221-222)
4. There is a typo in line 83: “angle” instead of “angel”.
Response: Thanks for the reviewer’s suggestion. We have replaced the “angel” to “angle” in the manuscript. (Page 4, line 132-133, 143-144)
Reviewer 3 Report
Comments and Suggestions for Authors
In this study, a miniaturized electrothermal-MEMS-based SS-OCT microscope was proposed for long-time handheld use in clinical settings. Nevertheless, some issues should be addressed:
1) The authors did not provide a comprehensive discussion of the implications and future directions of the research, limiting its overall impact and relevance in the field.
2) The article does not discuss potential limitations or biases in the study design, which is crucial for assessing the generalization of the findings.
3) The authors did not clearly state the methodology used in the research, making it challenging for readers to evaluate the validity and reliability of the results.
4) The authors should provide a thorough review of existing clinical imaging techniques, which is essential for understanding the limitations of the current methods
5) The authors should discuss some negative aspects or limitations of the proposed miniaturized electrothermal-MEMS-based SS-OCT microscope
6) Existing SS-OCT probes are still bulky and inconvenient for long-time handheld imaging
Author Response
We appreciate the valuable comments raised by the editor and reviewers. We have made a major revision of to improve the manuscript, addressed all the concerns, and highlighted the revisions of the revised manuscript in red.(The revised manuscript can be seen in the attachment)
1. The authors did not provide a comprehensive discussion of the implications and future directions of the research, limiting its overall impact and relevance in the field.
Response: Thanks for the reviewer’s suggestion. We added the comprehensive discussion of the implications and future directions of the research in the conclusion section. (Page 7, line 255-265)
2. The article does not discuss potential limitations or biases in the study design, which is crucial for assessing the generalization of the findings.
Response: Thanks for the reviewer’s suggestion. We added the discussion about our potential limitations or biases in the study design and future improvement plans in the discussion section. (Page 7, line 239-249)
3. The authors did not clearly state the methodology used in the research, making it challenging for readers to evaluate the validity and reliability of the results.
Response: Thanks for the reviewer’s suggestion. We add the detail instructions in section 2. (Page 2-3, line 79-89, 107-127)
4. The authors should provide a thorough review of existing clinical imaging techniques, which is essential for understanding the limitations of the current methods.
Response: Thanks for the reviewer’s suggestion. We have reviewed the existing clinical imaging techniques for human skin imaging in introduction. Besides, we have added some discussion about the advantages and limitations of existing imaging technologies both in introduction and discussion sections. (Page 1-2, line 27-43, 50-55, 60-64; Page 6, line 206-211)
5. The authors should discuss some negative aspects or limitations of the proposed miniaturized electrothermal-MEMS-based SS-OCT microscope.
Response: Thanks for the reviewer’s suggestion. We added the limitations and future improvement of the proposed miniaturized electrothermal-MEMS-based SS-OCT microscope in the discussion section. (Page 7, line 239-249)
6. Existing SS-OCT probes are still bulky and inconvenient for long-time handheld imaging.
Response: Thanks for the reviewer’s suggestion. In this study, we proposed MEMS-based SS-OCT microscopy only weighs about 25 grams and has a cylinder with a diameter of 15 mm and a length of 40 mm, showing a relative miniaturized volume. However, the appearance of probe is still need to be improved for more convenient hand-held imaging in the future, we added some explanation in the discussion section. (Page 3, line 98-100; Page 7, line 239-249)
Round 2
Reviewer 1 Report
Comments and Suggestions for Authors
I would suggest that the left part of Figure 1 be removed (the drawing of a PC etc). Also, check the right part of Figure 1 (there is a spelling error 'balanced detector'); it is not clear what is a zoomed-in part in the box.
Improve Figure 2.
The scale bar in Figure 5(c) is not visible.
Comments on the Quality of English LanguageThe quality of the English language is acceptable.
Author Response
We greatly appreciate the reviewer's valuable and insightful comments. We have carefully addressed all of the comments one by one. The revisions are highlighted in the revised manuscript in red. (The revised manuscript was uploaded as an attachment.)
1. I would suggest that the left part of Figure 1 be removed (the drawing of a PC etc). Also, check the right part of Figure 1 (there is a spelling error 'balanced detector'); it is not clear what is a zoomed-in part in the box.
Response: Thanks for pointing this out. We have corrected that on Page 3, Figure 1, lines 92-93, lines 103-107.
2. Improve Figure 2.
Response: Thanks for the reviewer’s suggestion. We have replaced Figure 2 (a) to a new picture with higher quality. Please refer to Page 4, Figure 2, lines 141-145.
3. The scale bar in Figure 5(c) is not visible.
Response: Thanks for pointing this out. We have corrected that on Page 6, Figure 5 (c), lines 192-199.
Reviewer 2 Report
Comments and Suggestions for Authors
The authors addressed properly most of my previous comments and the paper has clearly improved. There are still a few things that can be improved:
The authors now discuss why they selected a wavelength of 1300 nm and not 1060 nm, also available in SS-OCT technology. As it is, this discussion could be improved since the mention to the absorption and scattering properties of the skin is somewhat vague. It should be said that by selecting 1300 nm the authors are minimizing the scattering, which decreases strongly with the wavelength, although at the price of having more absorption than with 1060 nm, due to the absorption of water.
Author Response
We greatly appreciate the reviewer's valuable and insightful comments. We have carefully addressed all of the comments one by one. The revisions are highlighted in the revised manuscript in red.
1. The authors now discuss why they selected a wavelength of 1310 nm and not 1060 nm, also available in SS-OCT technology. As it is, this discussion could be improved since the mention to the absorption and scattering properties of the skin is somewhat vague. It should be said that by selecting 1310 nm the authors are minimizing the scattering, which decreases strongly with the wavelength, although at the price of having more absorption than with 1060 nm, due to the absorption of water.
Response: Thanks for the reviewer’s suggestion. We have discussed why we selected a wavelength of 1310 nm and not 1060 nm, modified and improved our previous viewpoint to “Especially for biological tissue imaging, considering that skin tissue has strong optical absorption and scattering characteristics, which will affect the penetration depth of laser. In order to maximize the imaging depth, the near-infrared band is generally used for imaging, such as 1060 nm or 1310 nm. With the wavelength increasing, the scattering decreases strongly. Therefore, we finally selected the 1310 nm wavelength to minimize the scattering even though 1310 nm has stronger absorption from the water than 1060 nm.”. Please refer to Page 6, lines 202-208.